# Section Optimization Design of a Flexible Cable-Bar Tensile Structure Based on Robustness

Lianmeng Chen [1,*], Yihong Zeng [1], Weifeng Gao [1], Yijie Liu [1] and Yiyi Zhou [2]

[1] College of Civil Engineering and Architecture, Wenzhou University, Wenzhou 325035, China; 20461542066@wzu.edu.cn (Y.Z.); 16461572113@wzu.edu.cn (W.G.); 194611571239@wzu.edu.cn (Y.L.)

[2] Changzhou Institute of Technology, College of Civil Engineering and Architecture, Changzhou 213002, China; zhouyy@czu.cn

[*] Correspondence: 00151034@wzu.edu.cn; Tel.: +86-13957790090

**Abstract:** As the current literature lacks effective nonlinear robustness evaluation method and optimal design theory of the structural robustness for flexible cable-bar tensile structure, this paper aimed to conduct further studies. Based on the $H_\infty$ theory, a fundamental robustness analysis method and a detailed calculation way through the combination of induction of $L_2$ performance criterion and random theory for nonlinear structural robustness quantitative evaluation method were proposed. Following this, a real Geiger cable dome structure was studied as its research object, and the influences of structural robustness of simultaneous changes of all elements section and changes of every kind of element section were analysed, respectively. Finally, the genetic algorithm was applied through MATLAB and ANSYS software to achieve optimal section layout, with the goal of minimizing structural quality on the condition that the structural robustness indicator keep less than that of the initial structure. The result revealed that the increase of the section of elements can effectively enhance structural robustness and the section changes of various elements showed different sensitivities to the influence of structural robustness. Meanwhile, structural quality can be effectively reduced by optimizing measures such as increasing the section of elements with significant effect on structural robustness and reducing the section of elements with minor effects on structural robustness, while the structural robustness indicator keeps less than that of the initial structure. The optimization reveals that quality was reduced by 42.5% in this paper.

**Keywords:** cable-bar tensile structures; structural robustness; section optimization; genetic algorithm

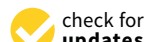



## 1. Introduction

Cable-bar tensile structure is a type of flexible tensile structure consisting of cables and bars. Due to the advantages of the high-strength cable and initial prestress, distribution can be adjusted to optimize the structural stiffness distribution. The structure is featured by great span, lightweight structure, graceful shape, low cost and other features, and has been applied in various engineering domains [1,2]. Meanwhile, due to the low redundancy of this type of structure, it is easy to produce continuous collapse when subject to overload, explosion and other accidental conditions [3–5]; that is, the chain reaction will be caused by the initial partial damage and will eventually lead to the overall structural collapse or a large-scale structural collapse that is disproportionate to the initial partial failure. In 2006, the Bad Reichenhall Skating Hall in Germany collapsed continuously after the damage scopes were transferred to the surrounding areas due to the failure of some nodes of the partial grids under the actions of overload [6] (as shown in Figure 1). In 2007, the dome structure of the Vancouver Winter Olympic Stadium collapsed due to fierce wind, rainstorm, and heavy snow (as shown in Figure 2). Yet, it is uneconomical to considerably enhance the design requirement of the structure just owing to unexpected interference or accidental overload. Therefore, it is a reasonable choice to optimize the design of the structure, making the structure insensitive to partial damage to avoid the collapse of the overall structure

caused by partial element damage or partial area failure and increase the structural capacity to resist the continuous collapse; that is, to enhance the structural robustness.

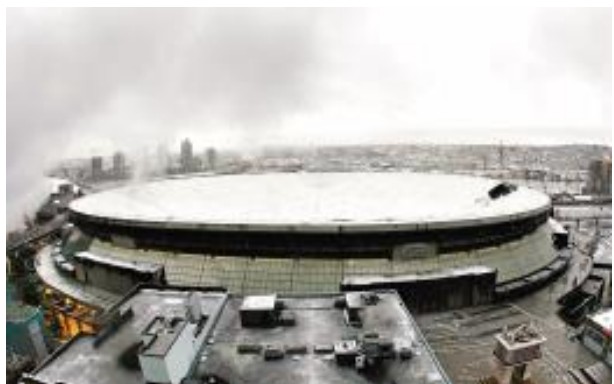

**Figure 1.** Vancouver Winter Olympic Stadium.

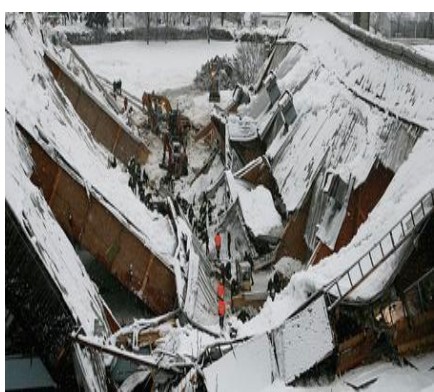

**Figure 2.** Bad Reichenhall Skating hall.

The concept of robustness was first presented in the control field in the 1960s as a way to solve the problem of unsteadiness in the optimization procedure. In the field of building structure, robustness was generally defined as the performance of a structure to withstand the consequences of damage that was disproportional to the cause [7,8], which emphasized the rationality of the distribution of topology and stiffness of the structure and required the absence of weakness or fatal defect within the structure. In 1968, the concept of robustness was firstly factored in the building structure after the 18-storey Ronan Point apartment in London collapsed in an explosion caused by a gas leak [9]. After the collapse of the World Trade Center resulted from the "9.11" terrorist attack in 2001, the continuous collapse and robustness of building structures have been gradually paid attention to by governments on a global scale, leading to the emphasis in their design regulations [10–12]. However, at present, these regulations mainly focus on qualitative description and indirect measures, which fail to provide a universal standard for building applications. Thus, a quantitative evaluation indicator is needed to conduct robustness optimization design and form design specifications. Meantime, the major research targets of structural robustness mainly involve masonry structure [13], frame structure [14], frame-core tube structure [15], bridge structure [16], underground structure [17], etc.

In recent years, a series of research studies have been performed in the domain of large-span structures, and the scopes of research are primarily concentrated on truss [18], beam [19], net frame [20], latticed shell [21], tension string structures [22], cable-supported grid structures [23], and so on. Yet, few studies have been done in relation to the robustness of flexible cable-bar tensile structures such as cable dome structures and spatial cable truss tension structures that are characterized with longer span, more diversified shapes, newer materials, higher sensitivity to construction errors. Additionally, flexible cable-bar

tensile structure is different from the rigid structure, it belongs to a geometrically variable system before it is prestressed and needed to be prestressed for stiffness to guarantee the bearing performance of the structure [24–26]. During the load process, cable-bar loosening and dysfunction might occur and turn the structure into a mechanism, or materials enter into a nonlinear state such as the plastic stage. Therefore, a flexible cable-bar tensile structure is featured with strong nonlinear characteristics, and the robustness evaluation method for a rigid structure is not applicable to this case. Furthermore, flexible cable-bar tensile structures are supported by the high-strength cable to bear load and ensure stiffness, and its elements bear high strain energy. Once these elements are destroyed, the consequence can be detrimental [27]. Hence, it is crucial to build a definite robustness-based quantitative assessment method for flexible cable-bar tensile structures, explore the robustness-based structural optimization design theory, and increase its structural ability to resist disproportionate damage and avoid eventual collapse.

In addition, current research of optimization of cable-bar tensile structure is centred on the fields of pre-stress optimization design, element section optimization design, structural shape optimization design and structural topology optimization design. Among them, most of the pre-stress optimization designs of spatial pretension structure aims to achieve the lowest pre-stress level as the optimal goal [28,29]. A majority of the optimal designs of element section [30] and structural shape [31] center around the obtainment of the lightest weight. In fact, the flexible cable-strut tensile structure is very light in weight and the use of steel weighs less than 30 kg/m$^2$. Hence, it carries little significance to conduct further research of only weight optimization on this basis and the structural performance should be taken into account at the same time, such as the structural robustness. However, optimal design theories based on structural robustness for nonlinear cable-bar tensile structures are also seldom considered presently. Therefore, it is of both great research value and engineering application prospect to build an optimization model for the cable-strut tensile structure based on structural robustness performance.

In view of the above, as the current literature lacks an effective nonlinear robustness evaluation method and optimal design theory of the structural robustness for flexible cable-bar tensile structure, based on the $H_\infty$ theory, a fundamental robustness analysis method and a detailed arithmetic calculation way through the combination of induction of $L_2$ performance criterion and random theory for nonlinear structural robustness quantitative evaluation method were firstly proposed. Following this, a real Geiger cable dome structure was studied as its research object, and the influences of structural robustness of simultaneous changes of all element sections and changes of every kind of element section were analysed, respectively. Finally, MATLAB programming and ANSYS software were used for genetic algorithm. The optimal goal was to achieve optimal section layout with the goal of minimizing structural quality on the condition that the structural robustness indicator keep less than that of the initial structure, and the result revealed that structural quality can be decreased by 42.5% on the condition that the structural robustness indicator be still less than that of the initial structure. Therefore, the implication of this study offered insights not only to theoretical study but also to building application.

## 2. Robustness Analysis Theory for Flexible Cable-Bar Tensile Structures

### 2.1. Fundamental Robustness Analysis Method Based on $H_\infty$ Theory

Presently, the definition of structural robustness is the capacity of a structure to avoid the continuous collapses resulted from unusual causes, and an evaluation indicator is needed to express the ratio of the damage consequences to the causes. Meanwhile, $H_\infty$ robust control theory is a well-developed theory in the control area and the $H_\infty$ norm is an important performance indicator in the robust control system. The state space model is usually adopted when describing the control system. From the perspective of input and output, the system is regarded as a mapping from input space to output space. The dynamic equation of the system is usually expressed in the form of state equation and output equation when using the state space model to describe a structural system. For a

linear time-invariant system $\sum_0$ with $r$ dimensional input and $m$ dimensional output, it can be expressed as:

$$\sum_0 : \begin{cases} \dot{x}(t) = A_0 x(t) + B_0 u(t) \\ y(t) = C_0 x(t) + D_0 u(t) \end{cases} \tag{1}$$

In this formula, $x(t) = [x_1(t)x_2(t) \cdots x_n(t)]^{\mathrm{T}}$ denotes the $n$ dimensional state vector of the system and is a group of vectors with the least number that can fully describe the time-domain behavior of the system. $u(t) = [u_1(t)u_2(t) \cdots u_r(t)]^{\mathrm{T}}$ denotes the input vector of the system, and $y(t) = [y_1(t)y_2(t) \cdots y_m(t)]^{\mathrm{T}}$ denotes the output vector of the system, which is adopted to depict the external state of the system. The characteristic matrices of system $A_0$, $B_0$, $C_0$, $D_0$ are the matrix of the $n \times n$ dimensional system matrix, $n \times r$ dimensional input matrix, $m \times n$ dimensional output matrix and $m \times r$ dimensional input–output coupling matrix, respectively.

Laplace transformation is carried out on both ends of Equation (1) of the system state space, as follows:

$$\sum_0 : \begin{cases} sX(s) - x(t_0) = A_0 X(s) + B_0 U(s) \\ Y(s) = C_0 X(s) + D_0 U(s) \end{cases} \tag{2}$$

In this formula, $U(s)$ and $Y(s)$ are the Laplace transform of $u(t)$ and $y(t)$ respectively, and $s$ is the complex frequency. In the formula (2), assuming initial conditions $x(t_0) = 0$ and $\dot{x}(t_0) = 0$, it can be obtained that:

$$Y(s) = \left\{ C_0(sI - A_0)^{-1} B_0 + D_0 \right\} U(s) \tag{3}$$

To define the transfer function of the $i$th output vector $y_i$ and the $j$th input vector $u_j$ in the frequency domain:

$$G_{0ij}(s) = \frac{Y_i(s)}{U_j(s)} \tag{4}$$

For the linear structure system, it can be obtained by superposition principle that:

$$Y(s) = G_0(s)U(s) \tag{5}$$

$$G_0(s) = C_0(sI - A_0)^{-1} B_0 + D_0 \tag{6}$$

In this formula, $I$ is the diagonal matrix, $G_0(s)$ is the system transfer matrix composed of $G_{0ij}(s)$. $G_0(s)$ describes the mapping relation between input variables $u(t)$ and output variables $y(t)$, and is an attribute of the structural system per se and is irrelevant to input variables $u(t)$.

The aforementioned fundamental idea of $H_\infty$ theory is to assume that the structural system is seen as a structural group containing uncertainties. If all objects in the structural group can meet the expected performance indicators through design, the actual structure will also meet the performance requirements. Therefore, the design of the structural robustness can actually be regarded as the process of optimizing the structural group, and the norm of the system transfer function $H_\infty$ is used as the optimization indicator. During the design process, the gain from input interference $w(t)$ to output $\Delta y(t)$ should be sufficiently small; that is, the $H_\infty$ norm of the system transfer function $G_{w\Delta y}(s)$ should be extremely small, which can be expressed as:

$$min\|G_{w\Delta y}(s)\|_\infty = \gamma_0 \tag{7}$$

$H_\infty$ optimizes the performance indicator in the worst state to enable the interference inhibition of the structural system to meet the requirements. Therefore, the evaluation indicator of structural robustness can be defined as:

$$I_R = \|G(s)_{w\Delta y}\|_\infty \tag{8}$$

For the flexible cable-bar tensile structures, input disturbances will affect structural properties due to its strong geometrical nonlinearity. Therefore, the further introduction of performance criteria $L_2$ is needed, and the input interference $w(t)$ and the output interference $\Delta y(t)$ belong to $L_2$. $W(t)$ and $\Delta Y(t)$ can be obtained by $w(t)$ and $\Delta y(t)$ through Laplace transform. The transfer function matrix of the structural system $G_{w\Delta y}(s) \in H_\infty$, the input interference vector $W(s) \in H_2^m$, the output interference vector $\Delta Y(s) \in H_2^m$, and the infinite norm of the matrix $G_{w\Delta y}(s)$ can be calculated via the induced norm $L_2$:

$$I_R = \|G_{w\Delta y}(s)\|_\infty = \sup_{\|W\|_2 \neq 0} \frac{\|\Delta Y(s)\|_2}{\|W(s)\|_2} \tag{9}$$

For any function, $F, G \in L_2^n$, and the inner product of functions can be displayed as:

$$\langle F, G \rangle = \int_0^\infty F^T(t)G(t)dt \tag{10}$$

$w(t)$ and $\Delta y(t)$ are the inverse Laplace transform of $W(s)$ and $\Delta Y(s)$, respectively. In accordance with the definition of the inner product of functions,

$$\|W(s)\|_2 = \|w(s)\|_2 \tag{11}$$

$$\|\Delta Y(s)\|_2 = \|\Delta y(t)\|_2 \tag{12}$$

By substituting Equations (11) and (12) into Equation (9), the robustness of nonlinear structural system can be displayed as:

$$I_R = \|G_{w\Delta y}(s)\|_\infty = \sup_{\|w\|_2 \neq 0} \frac{\|\Delta y(t)\|_2}{\|w(t)\|_2} \tag{13}$$

Among them: $\|(\cdot)\|_2 = \left(\int_0^\infty \|(\cdot)\|^2 dt\right)^{1/2} = \left(\int_0^\infty (\cdot)^T Q(\cdot)dt\right)^{1/2}$, Q was a weighed matrix.

According to Equation (13), the quantitative robustness evaluation indicator $I_R$ of the nonlinear structure can be calculated. Since the indicator $I_R$ indicates the sensitivity of the structural performance to resist the unexpected input interference, the greater the indicator $I_R$, the weaker the resistance of the overall structure to the input interference, namely, the weaker the structural robustness, and vice versa.

*2.2. Calculation Method of Robustness Analysis for Tensile Structures Based on Random Theory*

In this paper, input interference and output response were quantified as the node load interference input vector and the corresponding node displacement output vector. Then, the structural robustness indicator could be established by calculating the proportion of the output node displacement and the input interference load. The input interference $w(t)$ consisted of substantial independent random factors and could be reckoned as a normal distribution. According to the principle of normal distribution function in probability theory, the chance of interference occurrence exceeded 99.74% in the interval $(\mu - 3\sigma, \ \mu + 3\sigma)$, in which $\mu$ was the average value and $\sigma$ was the standard deviation. Therefore, although the range of normal variables was $(-\infty, +\infty)$, it was certain that the interference load occurred within the range $(\mu - 3\sigma, \ \mu + 3\sigma)$. Based on the theory, the specific calculating method for the robustness of cable-bar tensile structure comprised the following steps.

(1) The total combined loads of cable-bar tensile structure included conventional load $F_0$ and external interference load $w(t)$. The conventional loads included static load and dynamic load. The interference load obeyed the normal distribution, the distribution interval was $(-3var, \ 3var)$, and the area was divided into $m$ sections. The

ratio between the interference load and the conventional load in the interval $k$ was as follows:

$$\alpha(k) = \pm \frac{6k \cdot var}{m}, \ k = 1, 2, 3, \cdots, m/2 \tag{14}$$

In the formula, $var$ was the coefficient of variation, which was set as 0.005 in this paper, then the interference load within the interval could be calculated as follows:

$$w_k(t) = F_0 \cdot \alpha(k), \ k = 1, 2, 3, \cdots, m/2 \tag{15}$$

The combined load $F_k$ in each sub-section was the sum of the conventional load $F_0$ and the interference load $w_k(t)$, which was calculated as follows:

$$F_k = F_0 + w_k(t), \ k = 1, 2, 3, \cdots, m/2 \tag{16}$$

$F_k$ was subjected to the normal distribution of parameters $\mu$ and $\sigma$ in the interval $(\mu - 3\sigma, \ \mu + 3\sigma)$, in which $\mu$ was the average value and $\sigma$ was the standard deviation. Then, the following formula could be obtained:

$$\sigma = \mu \cdot var \tag{17}$$

To ensure the efficiency and accuracy of the calculation results, $m$ was 100 in this paper. Meanwhile, since the interference load $w(t)$ caused both positive and negative occurrence, and the occurrence probability of the same absolute value of positive and negative interference load was equal, the value of the probability interval number is 50/50.

(2) The nodal displacement vector under the conventional load $F_0$ and the combined load $F_k$ were $y$ and $y_k$, respectively, which could be calculated by software ANSYS. According to the interval divided by the combined load in step (1), the probability distribution function of the combined load in interval $k$ was calculated and used as the weight coefficient. The formula was as follows:

$$Q(k) = \begin{cases} \int_{\frac{6k-3}{m}}^{\frac{6k+3}{m}} \frac{1}{\sqrt{2\pi}} e^{-\frac{t^2}{2}} dt, & k = 1 \sim \frac{m}{2} - 1 \\ 1 - \int_{-\infty}^{3 - \frac{3}{2k}} \frac{1}{\sqrt{2\pi}} e^{-\frac{t^2}{2}} dt, & k = \frac{m}{2} \end{cases} \tag{18}$$

(3) For robustness indicator $I_{Rk}$ of any interval $k$, the robustness indicator was the ratio of node displacement under the interval interference load and the interference load in the interval, that is:

$$I_{Rk} = \sum_{i=1}^{n} \frac{\sqrt{(u'_{kxi} - u_{xi})^2 + (u'_{kyi} - u_{yi})^2 + (u'_{kzi} - u_{zi})^2}}{F_0 \cdot \alpha(k)}, \ k = 1, 2, 3, \cdots, m/2 \tag{19}$$

In this formula, robustness indicator $I_{Rk}$ was generated by the interference load in interval $k$; $n$ was the total number of structure nodes; $u_{xi}$, $u_{yi}$, $u_{zi}$ were the node displacement components of the node $i$ under the conventional load in the directions of $x$, $y$, $z$, respectively; $u'_{kxi}$, $u'_{kyi}$, $u'_{kzi}$ were the node displacement components of the node $i$ under the combined load in interval $k$ in the three directions $x$, $y$, $z$; $\alpha(k)$ was the ratio between the interference load $w_k(t)$ and the conventional load $F_0$ in the interval $k$.

(4) Robustness indicator $I_R$ of the structure within the range of the combined load was calculated. Based on the structural robustness in the specified combined load interval $k$ obtained in step (3), the weighted sum was used to obtain the structural

robustness indicator in the normal distribution interval, and the calculation formula was as follows:

$$I_R = \sum_{k=1}^{m/2} \sum_{i=1}^{n} \frac{Q(k) \cdot \sqrt{(u'_{kxi} - u_{xi})^2 + (u'_{kyi} - u_{yi})^2 + (u'_{kzi} - u_{zi})^2}}{F_0 \cdot \alpha(k)}, \ k = 1, 2, 3, \cdots, m/2 \quad (20)$$

## 3. Influence of Element Sections on Structural Robustness

The Yiqi National Fitness Sports Centre in Inner Mongolia was taken as a case in point in this study [32] (as shown in Figure 3). The Centre was a Geiger cable dome structure, which was a typical flexible cable-bar tensile structure, and its span was 71.2 m, rise 5.5 m, and rise–span ratio about 1/13. This cable dome consisted of twenty pieces of symmetrical cable-bar units, and each unit comprised cables and bars. The cables consisted of three kinds of tension cables, such as diagonal cables, ridge cables and hoop cables. Diagonal cables consisted of outer diagonal cables (abbreviated as DC1), middle diagonal cables (abbreviated as DC2) and inner diagonal cables (abbreviated as DC3). Ridge cables comprised outer ridge cables (abbreviated as RC1), middle ridge cables (abbreviated as RC2) and inner ridge cables (abbreviated as RC3). Hoop cables consisted of outer hoop cables (abbreviated as HC1), middle hoop cables (abbreviated as HC2), top hoop cables in inner ring (abbreviated as THC) and lower hoop cables in inner ring (abbreviated as LHC). The compression bars consisted of outer bars (abbreviated as WG1), middle bars (abbreviated as WG2), and inner bars (abbreviated as WG3). The design parameters of each element and initial pre-stress of the overall structure are illustrated in Table 1, and the elastic modulus of cable and bar were 160 GPa and 206 GPa, respectively.

**Table 1.** Initial pre-stress and section parameters of structural elements.

| Element Name | DC1 | DC2 | DC3 | RC1 | RC2 | RC3 | WG1 | WG2 | WG3 | HC1 | HC2 | THC | LHC |
|---|---|---|---|---|---|---|---|---|---|---|---|---|---|
| Pre-stress (KN) | 466.6 | 208 | 105.9 | 682.2 | 473.1 | 370 | −158 | −70.4 | −36.2 | 1403.2 | 625.7 | 1190.1 | 305.3 |
| Initial area (mm$^2$) | 2488 | 853 | 605 | 1844 | 1361 | 853 | 7804 | 4674 | 4674 | 7466 | 3318 | 3318 | 3318 |
| Optimized Area (mm$^2$) | 1812 | 452 | 81 | 1050 | 1663 | 1568 | 1232 | 443 | 213 | 13,800 | 2650 | 3685 | 233 |
| Area change ratio | −27% | −47% | −87% | −43% | 22% | 84% | −84% | −91% | −95% | 85% | −20% | 11% | −93% |

### 3.1. Influences of Simultaneous Changes of All Sections of the Elements on Structural Robustness

When all sections of the elements were enlarged or diminished simultaneously, structural robustness changes were shown in Table 2. The following findings were observed: (1) the indicator $I_R$ decreased with the simultaneous increase of all element sections; namely, structural robustness increased with the increase of the element section. (2) For the same kind of elements, the reduction of element section area had a more significant effect than the increase of the same element section area on structural robustness. For example, in comparison with the initial element section area, when the section area was reduced or enlarged by 40%, the structural robustness indicator increased or decreased by 66.8% and 28.6%, respectively.

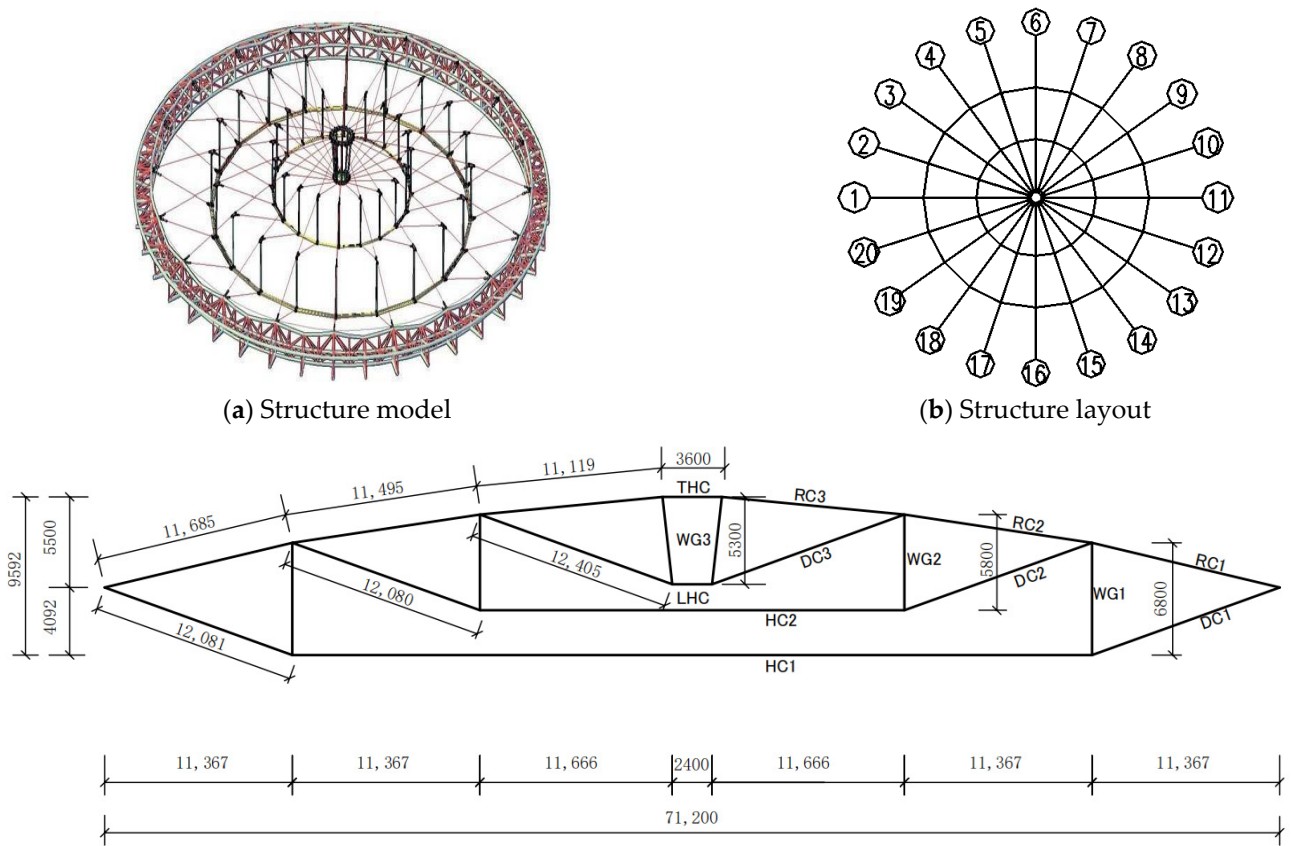

(**a**) Structure model

(**b**) Structure layout

(**c**) Structure profile and element size (mm)

**Figure 3.** A cable-bar tensile structure in Inner Mongolia.

**Table 2.** Robustness indicator under various sections (unit: $10^{-4}$ m/KN).

| Section Area | 0.4 A | 0.6 A | 0.8 A | 1.0 A | 1.2 A | 1.4 A | 1.6 A |
|---|---|---|---|---|---|---|---|
| Robustness indicator | 9.252 | 6.165 | 4.622 | 3.697 | 3.081 | 2.640 | 2.310 |

*3.2. Influences of Changes of Each Kind of Element Sections on Structural Robustness*

The influences of the changes of each kind of element sections on structural robustness were further evaluated. The robustness of the initial element section area of each kind of elements multiplied by various amplification and reduction coefficients were calculated. The results are shown in Table 3, in which $I_R$ stood for the robustness indicator when each kind of element was under the various element sections distributions, and $I_{R0}$ represented the robustness indicator of the initial structure.

It is shown in Table 3 that: (1) The element section changes of various kinds of elements showed various sensitivities to structural robustness. Outer hoop cable (HC1) was the most sensitive element of all and following this were inner ridge cable (RC3), outer diagonal cable (DC1) and middle ridge cable (RC2). (2) The section changes of bars, including WG1, WG2, and WG3, together with low hoop cable in inner ring (LHC) and inner ridge cable (RC3), were negligible, given the fact they exerted little influence on structural robustness. (3) The sensitivities of section changes of middle diagonal cable (DC2), outer ridge cable (RC1), middle hoop cable (HC2) and top hoop cable in inner ring (THC) to structural robustness stood between the two abovementioned kinds of elements.

**Table 3.** Structural robustness indicator under various section-area of each kind of elements.

| Elements | Section Area | 0.4 A | 0.6 A | 0.8 A | 1.0 A | 1.2 A | 1.4 A | 1.6 A |
|---|---|---|---|---|---|---|---|---|
| DC1 | $I_R$ | 4.470 | 4.060 | 3.837 | 3.697 | 3.601 | 3.530 | 3.477 |
|  | $(I_R - I_{R0})/I_{R0}$ | 20.9% | 9.8% | 3.8% | 0% | −2.6% | −4.5% | −6.0% |
| DC2 | $I_R$ | 4.058 | 3.860 | 3.759 | 3.697 | 3.655 | 3.625 | 3.603 |
|  | $(I_R - I_{R0})/I_{R0}$ | 9.8% | 4.4% | 1.7% | 0% | −1.1% | −1.9% | −2.5% |
| DC3 | $I_R$ | 3.660 | 3.679 | 3.690 | 3.697 | 3.702 | 3.706 | 3.708 |
|  | $(I_R - I_{R0})/I_{R0}$ | −1.0% | −0.5% | −0.2% | 0% | 0.1% | 0.2% | 0.3% |
| RC1 | $I_R$ | 3.864 | 3.783 | 3.732 | 3.697 | 3.672 | 3.652 | 3.636 |
|  | $(I_R - I_{R0})/I_{R0}$ | 4.5% | 2.3% | 0.9% | 0% | −0.7% | −1.2% | −1.6% |
| RC2 | $I_R$ | 4.338 | 4.013 | 3.823 | 3.697 | 3.608 | 3.542 | 3.491 |
|  | $(I_R - I_{R0})/I_{R0}$ | 17.3% | 8.5% | 3.4% | 0% | −2.4% | −4.2% | −5.6% |
| RC3 | $I_R$ | 4.735 | 4.209 | 3.901 | 3.697 | 3.553 | 3.445 | 3.361 |
|  | $(I_R - I_{R0})/I_{R0}$ | 28.1% | 13.8% | 5.5% | 0% | −3.9% | −6.8% | −9.1% |
| WG1 | $I_R$ | 3.712 | 3.704 | 3.700 | 3.697 | 3.696 | 3.694 | 3.693 |
|  | $(I_R - I_{R0})/I_{R0}$ | 0.4% | 0.2% | 0.1% | 0% | 0% | −0.1% | −0.1% |
| WG2 | $I_R$ | 3.699 | 3.698 | 3.697 | 3.697 | 3.697 | 3.697 | 3.697 |
|  | $(I_R - I_{R0})/I_{R0}$ | 0.1% | 0% | 0% | 0% | 0% | 0% | 0% |
| WG3 | $I_R$ | 3.697 | 3.697 | 3.697 | 3.697 | 3.697 | 3.698 | 3.698 |
|  | $(I_R - I_{R0})/I_{R0}$ | 0% | 0% | 0% | 0% | 0% | 0% | 0% |
| HC1 | $I_R$ | 5.045 | 4.356 | 3.958 | 3.697 | 3.512 | 3.373 | 3.265 |
|  | $(I_R - I_{R0})/I_{R0}$ | 36.5% | 17.8% | 7.1% | 0% | −5.3% | −8.8% | −11.7% |
| HC2 | $I_R$ | 3.975 | 3.823 | 3.745 | 3.697 | 3.665 | 3.642 | 3.625 |
|  | $(I_R - I_{R0})/I_{R0}$ | 7.5% | 3.4% | 1.3% | 0% | −0.9% | −1.5% | −1.9% |
| THC | $I_R$ | 3.861 | 3.771 | 3.725 | 3.697 | 3.678 | 3.665 | 3.655 |
|  | $(I_R - I_{R0})/I_{R0}$ | 4.4% | 2.0% | 0.8% | 0% | −0.5% | −0.9% | −1.1% |
| LHC | $I_R$ | 3.696 | 3.696 | 3.697 | 3.697 | 3.697 | 3.698 | 3.698 |
|  | $(I_R - I_{R0})/I_{R0}$ | 0% | 0% | 0% | 0% | 0% | 0% | 0% |

## 4. Optimal Design of Element Section Based on Structural Robustness

The aforementioned analysis indicated that the section changes of different kinds of elements displayed different degrees of sensitiveness and efficiencies to structural robustness. In this section, further optimization design based on genetic algorithm was conducted. The main program of genetic algorithm was written in numerical analysis software MATLAB, and structural modeling, node displacement and robustness indicator $I_R$ IRwere calculated in ANSYS software. Then, ANSYS software was called for by MATLAB software automatically and the two software could cross-read the data. The specific process is shown below:

(1) Optimization parameters were set in MATLAB, including population size, encoding string length, crossover and mutation probability and number of evolutionary iterations. In this paper, the population size was set as 40, encoding string length was 52, crossover probability was 0.8, mutation probability was 0.2, and the number of evolutionary iterations was 200. Then, the optimization variable was selected, the real value range of the variable was calculated, and the initial population was generated by binary coding.

(2) The initial population that was generated by MATLAB was imported into ANSYS software. Meanwhile, the structural robustness indicator $I_R$ that was represented by each individual of the population was calculated. The number of structural robustness indicators was equal to the population size.

(3) The structural robustness indicator $I_R$ that was calculated by ANSYS software was imported into MATLAB and the reciprocal of the robustness indicator $1/I_R$ was taken as the fitness function, the sequencing was carried out, and the process of replication, crossover and mutation was selected after the extraction of the best individual of the initial population.

(4) Steps (2)~(3) were repeated for cyclic iterative calculation, the best individual of the offspring was extracted until the optimal value of robustness was obtained.

The most optimal algorithm for structural robustness was explored and the optimal section layout within a controlled sectional scope of all kinds of elements was searched with the goal of minimizing structural quality. Meanwhile, the structural robustness indicator being less than that of the initial structure was ensured, that is, the structural robustness should remain more superior than that of the initial structure, the mathematical optimization model could be expressed as Equation (21).

$$\begin{cases} \underset{A}{min}M \;=\; \sum_{i=1}^{13} \rho L_i A_i \\ s.t.\; I_R \;\leq\; I_{R0} \\ A_{imin} \;\leq\; A_i \;\leq\; A_{imax} \end{cases} \tag{21}$$

In this formula, the change range of the element section area was to ensure that the stress of all kinds of elements under the combined action of conventional load and interference load did not exceed the material yield strength of the lower limit, and the upper limit was 2.5 times the area of each element of the initial structure model.

The optimal process and results are shown in Figure 4. With the increasing number of iterations, structural quality keeps decreasing, and finally reached a plateau after 130 iterations. The optimal value of quality was 1230.53 kg, which was 42.5% lower than the initial structural model's quality of 2138.96 kg. Meanwhile, the robustness indicator of the optimal structure was $3.694 \times 10^{-4}$ m/KN, which was less than that of the initial structural model's robustness indicator of $3.697 \times 10^{-4}$ m/KN. The distribution of sections of each element after optimization is shown in Table 1.

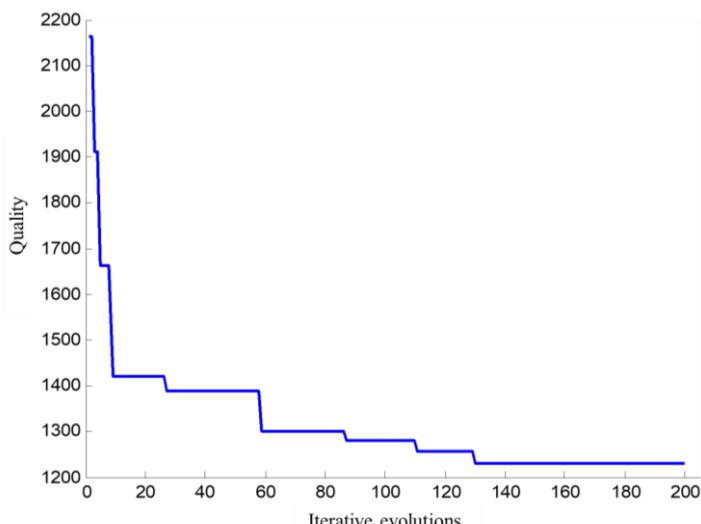

**Figure 4.** Iterative graph of structural quality optimization.

As can be seen from Table 1: (1) After the optimization of the structure, some sections of tension cable elements increased and other tension cables sections decreased, whereas the sections of compression bars elements all decreased; (2) From the degrees of section change, the sections of outer hoop cables and inner ridge cables, the elements with the most noticeable changes were increased by more than 80%. Meanwhile, the sections of middle ridge cables and top hoop cables in the inner ring were also increased by 22% and 11%, respectively. With reference to the previous analysis, the section changes of outer hoop cables, inner ridge cables, and middle ridge cables were all noticeably sensitive to the influence of structural robustness; (3) Section reductions were more evident in the case of bars, low hoop cables in inner ring and inner diagonal cables. The section of inner bars, the element with the most significant reduction, was reduced by 95%. Based on the

aforementioned analysis, the influences of section changes of bars, low hoop cables in inner ring and inner diagonal cables on structural robustness were insensitive. In conclusion, the sections of elements with significant influence on structural robustness were increased after optimization, while the sections of elements without noticeable influence were decreased after optimization.

## 5. Conclusions

Based on the $H_\infty$ theory, a fundamental robustness analysis method and a detailed arithmetic calculation way through the combination of induction of $L_2$ performance criterion and random theory for nonlinear structural robustness quantitative evaluation method were proposed. Following this, a real Geiger cable dome structure was used as its research object, and the influences of structural robustness of simultaneous changes of all elements sections and changes of every kind of element section were analysed, respectively. Finally, the genetic algorithm was applied through MATLAB and ANSYS software to achieve optimal section layout with the goal of minimizing structural quality on the condition that the structural robustness indicator maintain less than the robustness indicator of the initial structure. The results revealed that: (1) The increase of section of members could effectively enhance structural robustness; (2) The section changes of various kinds of elements showed different sensitivities to structural robustness. In this pager, outer hoop cable was the most sensitive element of all, and following this were inner ridge cable, outer diagonal cable, and middle ridge cable. The section changes of bar, low hoop cable in inner ring, and inner ridge cable were negligible, and they exerted little influence on structural robustness; (3) Under the condition that the structural robustness indicator is less than the robustness indicator of the initial structure, structural quality can be effectively reduced by optimizing measures such as increasing the area of elements with a significant effect on structural robustness and reducing the area of elements with minor effect on structural robustness. The optimization reveals that quality was reduced by 42.5% in this paper.

**Author Contributions:** Conceptualization, L.C.; methodology, L.C.; software, Y.L.; validation, Y.Z. (Yihong Zeng), W.G.(Weifeng Gao); resources, L.C. and Y.Z. (Yiyi Zhou); first draft of manuscript, L.C.; proofreading and revision, L.C.; supervision, L.C.; project administration, L.C. All authors have read and agreed to the published version of the manuscript.

**Funding:** This research was funded by the National Natural Science Foundation of China, grant number 51578422, and the National Natural Science Foundation of China, grant number 51678082.

**Acknowledgments:** The authors are grateful to the Natural Science Foundation of China for supporting this work (Grants No. 51578422, 51678082).

**Conflicts of Interest:** The authors declare no conflict of interest.

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
