# Peer review of "Section Optimization Design of a Flexible Cable-Bar Tensile Structure Based on Robustness"

_applsci, doi:10.3390/app11198816_

Round 1

Reviewer 1 Report

This paper is a piece of well-elaborated scientific research work oriented to propose a nonlinear structural robustness quantitative evaluation method and a section optimization design theory based on the structural robustness for flexible cable-bar tensile structures. The aforementioned study is both intriguing and significant for scientific research and practical application. The submitted manuscript can be a good paper and thus should be published. However, the following comments are suggested for consideration prior to the publication of this paper:

  • Please explain the differences between the robustness evaluation method proposed in this paper and the other robustness analysis theories proposed by other researchers in section 1.
  • In this paper, input interference and output response were quantified as the node load interference input vector and the corresponding node displacement output vector. Any other interference input vector and output vector can be selected?
  • During the introduction section, some related studies on tensile structures and optimizations published in recent years can be cited and briefly mentioned. For instance: Automated element grouping and self-stress identification of tensegrities, Engineering Computations, 2015;  Symmetry representations and elastic redundancy for members of tensegrity structures, Composite Structures, 2018; Geometric design classification of kirigami-inspired metastructures and metamaterials, Structures, 2021; Stiffness contributions of tension structures evaluated from the levels of components and symmetry subspaces, Mechanics Research Communications, 2019; Force identification of prestressed pin-jointed structures, Computers & Structures, 2011.
  • What does the symbol s means in G(s).
  • The optimization rate is 42.5% in this paper. This reviewer is quite interested in this point: any other measures can increase the rate?

Reviewer 2 Report

The paper reports a detailed and systematic study on the section optimization design of a flexible cable-bar tensile structure based on robustness. The methods and data presented in this paper is well understood and modeled in detail. Overall the manuscript is interesting, some minor comments can be found below.

  • In introduction (line 74) authors mentioned some studies have been done in relation to the robustness of flexible cable-bar tensile structure. In my opinion those studies should be cited.
  • In section 3,  the Yiqi national fitness sports center in inner Mongolia was taken as a case study, authors should mention why this particular structure was taken as a case study? Additionally mention the source they received the data of this structure from.
  • In figure 3 (c), the line weight of structural elements and dimension bar is same, this makes the figure hard to read, it is recommended to use the light line weight for dimension bars. Also, the units are not provided, if not possible to add in figure due to space, then mention the units in figure caption.
